# Paired assistance and poverty alleviation: Experience and evidence from China

**Quanzhong Wang**[1]*, **Zhongbao Tian**[2], **Sai Zhu**[1]

**1** College of Economics and Management, Anqing Normal University, Anqing, China, **2** College of Economics and Management, Nanjing Agricultural University Nanjing, Nanjing, China

* catzitt@sina.com

**Data Availability Statement:** The data used in this paper are from the information disclosure directory/annual report of the official website of the People's Government of W County, Lu 'an City, Anhui Province, and the poverty alleviation

## Abstract

This paper used the micro panel data from 2016 to 2019 of 2031 registered poor households in B Town, W County, Lu'an City of Anhui Province in China to analyze the diversified patterns and poverty alleviation effect of paired assistance based on the PSM-DID model. The empirical results show that paired assistance provided by social forces can significantly contribute to the poverty alleviation of poor households, promoting the poverty alleviation rate by 7.8%, which can be concluded through sample matching and control of relevant variables. Furthermore, based on the subsample of poor households with social assistance, we found that external social assistance subject to paired assistance can significantly improve the poverty alleviation rate of poor households by 14.26%, mainly hung on their economic base and strength of poverty alleviation.

## 1. Introduction

Poverty is a challenge that many countries face in their development (Magombeyi & Odhiambo, 2018; Callixte *et al.*, 2020) [1, 2]. In the past 40 years, China has reduced nearly 800 million people from the poverty, accounting for more than 75% of the global population of poverty alleviation during the same period. As shown in Fig 1, China's poverty alleviation has made great contributions to the world's poverty reduction. Therefore, a summary of China's experience in poverty alleviation may be helpful for other countries.

The mechanisms through which social forces contribute to poverty alleviation, particularly in the context of China's experience, have not been sufficiently studied. This research aims to address this gap by exploring the following research questions: What are the mechanisms through which paired assistance contributes to poverty alleviation in China? How do different types of social forces impact poverty alleviation? What are the poverty alleviation effects of governmental assistance compared to social assistance? To answer these questions, this study provides a framework on the composition and operation of social forces involved in pairing support. Then, by investigating the data of all poor households in a township, the PSM-DID model is utilized to analyze the impact of government assistance and social assistance on poor households to get out of poverty. Furthermore, the impact of external assistance type and internal assistance type in social assistance on poor households to get out of poverty, so as to

statistical statements of B Town from 2016 to 2019 published, disclosed and archived on the website of the county Poverty Alleviation and Development Office.

**Funding:** This research was funded by the Anhui Philosophy and Social Science Planning Project: "Research on Poverty Reduction Mechanism of Supported Industries from the Perspective of double-win for Villages and Households" (grant number AHSKQ2019D112). Funders played a major role in the study in terms of data collection.

**Competing interests:** The authors have declared that no competing interests exist.

compare the impact of different types of assistance subjects on poor households to get out of poverty. This will help build a sustainable poverty reduction ecosystem to prevent poor households from returning to poverty and promote their income growth.

## 2. Literature review

### 2.1 Theoretical support

The theoretical support of this study mainly comes from the theory of sustainable development and the theory of new public management. The new public management theory advocates the introduction of economic thinking into government administration, the government as the decision maker and leader, the citizen's demand as the orientation, the introduction of competition mechanism, and the emphasis on results rather than process. In poverty alleviation work, the government acts as a guide to identify and meet the needs of the poor, guide social resources to participate, improve the efficiency of government work through performance assessment, and promote the success rate of poverty alleviation work. The new public management theory provides valuable reference and guidance for the global poverty alleviation work.

In its poverty alleviation work, China has fully adopted the concept of sustainable development. On the one hand, it has increased social security funds for poverty alleviation, increased investment in rural infrastructure and public services, and improved the poverty alleviation environment for poor households. On the other hand, under the circumstances of respecting the will of the poor households and taking full account of local resource factors, local natural resources should be appropriately developed according to local conditions, local characteristic industries should be built, endogenous impetus should be generated, and the sustainable development ability of the poor areas should be improved, so as to achieve coordinated development in various aspects such as society, economy, culture, resources, environment and life to a certain extent.

### 2.2 Literature review

As regard to poverty alleviation plans and guidelines, China has issued a series of policies to encourage and mobilize social forces or groups to participate in poverty alleviation and development. In April 1994, the State Council issued the *National Poverty Alleviation Plan (1994–2000)*, which defines the specific ways and means to mobilize party and government

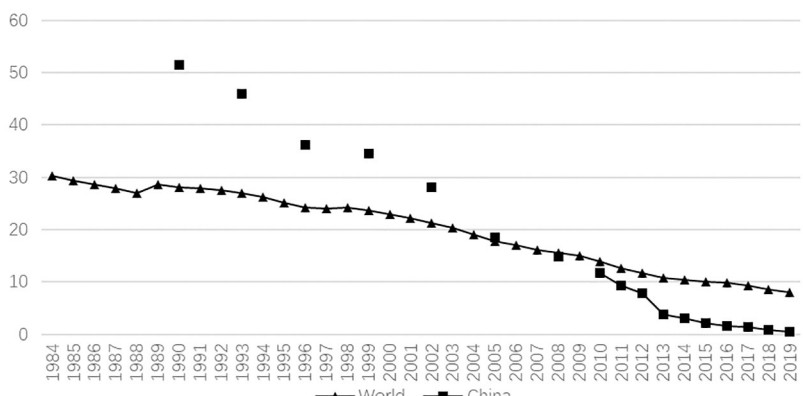

**Fig 1. Poverty incidence rate (data specification; the poverty rate data for China and the world presented in this paper are drawn from the World Bank's open database indicator 'poverty gap at $3.65 a day (2017 PPP) (%)).**

organs, enterprises and institutions in good conditions, democratic parties, trade union organizations and scientific research units at all levels to take advantage of their strengths to participate in poverty alleviation work actively. After entering the 21st century, the *Outline of China's Rural Poverty Alleviation and Development (2001–2010)* continued to include the mobilization of all sectors of society to help the development and construction of poverty-stricken areas as a policy guarantee, and provide tax concessions for donated funds. The subsequent program, the *China Rural Poverty Alleviation and Development Program (2011–2020)* gives more evident arrangements for social poverty alleviation, such as encouraging, guiding, supporting and helping various non-public enterprises and social organizations to undertake the task of targeted poverty alleviation, carrying out regional pairing work and mobilizing enterprises and social sectors to participate in poverty alleviation (Kong, 2018) [3]. Thanks to the above arrangements and incentives of a series of anti-poverty policies and initiatives, groups, teams and organizational forms of social forces participating in poverty alleviation and development work gradually grow and become increasingly involved. Relatively diversified patterns of subjects of paired assistance have gradually formed, effectively reducing the incidence of rural poverty and improving the efficiency of poverty alleviation (Yang *et al*., 2021) [4]. But the emphasis on social forces as an essential complementary tool for poverty alleviation in China, the existing research at home and abroad has not yet paid enough attention to the mechanisms and poverty alleviation effects of social forces' participation in poverty alleviation. Relevant studies have contrarily focused on the theoretical elaboration of social and governmental anti-poverty governance systems, role mechanisms and case studies (Hartmann *et al*., 2019) [5]; for example, Zhuang *et al*. (2015) point out that it's necessary to break the traditional government model of poverty alleviation and build a five-pronged poverty governance model of government–market–society–community–farmers [6]. According to Chen *et al*. (2017), the higher the level of paired support, the more sustainable the support measures for poor households, which can significantly increase recognition among those helped [7]. Leskosek (2012) found that social capital significantly mitigated illness-related poverty among households and that bridge-type social capital, associated with people of different socioeconomic statuses, had a more substantial impact on poverty alleviation than knot-type social capital, which is related to intra-family interactions [8]. In the context of COVID-19, recent studies had shown that poverty had worsened in many parts of the globe as a result of the economic downturn and unemployment caused by the epidemic (Mohsin *et al*., 2020; Muhammad, 2021; David *et al*., 2021; Sohail, 2022; Muhammad *et al*., 2022;) [9–12]. At the same time, poverty alleviation strategies should be integrated with sustainable development goals to achieve social justice, environmental protection and economic stability. For example, we can explore how to realize the dual goals of poverty alleviation and sustainable development by upgrading education and medical services, improving agricultural technology and promoting a green economy(Zhang, 2011; Lamini et al., 2021; Arsalan et al., 2022; Arsalan et al., 2022; Muhammad et al., 2023; Muhammad et al., 2023;) [13–18].

It is worth noting that some literature describes the poverty alleviation practices and mechanisms of some specific type of subject and organization of social assistance (such as education assistance, village task force, social work and cultural poverty alleviation by the non-profit bookstore) (Kumi & Copestake, 2022; Forkuor & Korah, 2022) [19, 20].

In conclusion, while significant progress has been made in reducing poverty, we still have large gaps in some areas, particularly in the role and impact of social forces, the nuances of matching levels of support, the role of different types of social capital, and the adaptation of strategies to new global challenges such as COVID-19 and the Sustainable Development Goals.

## 3. Conceptual framework

### 3.1 The China's framework for composition and operation of social forces involved in pairing support

This section outlines a conceptual framework to understand the China's framework for composition and operation of social forces involved in pairing support. Combined with the observation of poverty alleviation work in China, it is found that the emergence of diversified pairing help subjects can essentially be regarded as a problem of realizing the optimization of matching between poor households and pairing help subjects, help instruments and programs, with a view to reducing the misallocation of poverty alleviation target groups and poverty alleviation resources. To sort out the operational mechanism of differentiated pairing subjects' participation in poverty governance, it is necessary to have a clear definition of the functions and categories of different participating subjects.

The basic framework of China's precise poverty alleviation and support work is government-led, and the government helpers and social helpers to be distinguished in the latter part of the paper are all under the leadership of the Central Poverty Alleviation Work Office to accomplish poverty alleviation work or tasks. Particularly, in the mode of pairing help for poor households, the government and social help forces are not strictly separated or have two independent systems, the most obvious of which is that the data and information on poor counties, poor villages and poor households identified by the Central Poverty Alleviation Work Office are important bases for various social help forces to invest resources in helping them, while for some poverty alleviation means where social help forces are under pressure or beyond their resource allocation capacity, the government is still the main force. For some means of poverty alleviation that are under pressure or beyond its resource allocation capacity, the government is still the main helper, such as investment in rural infrastructure construction, special funds for education and poverty alleviation, and social assistance (underwriting) for the five-guarantee households(The five-guarantee households include the elderly, minors and disabled people in rural areas who are incapable of working, have no economic resources, and have no support (for the elderly) to support (for minors) the obligors, or have but are incapable of doing so, low-income households and special hardship groups).

On the above basic poverty alleviation system, it is necessary to define the compositions of the governmental and social assistance subjects. The governmental assistance subjects refer to the county-level leading group for poverty alleviation and development (poverty alleviation office), township poverty alleviation workstations, township governments and village (residents) committees (including resident working teams of poverty alleviation). On the contrary, the social assistance subjects refer to non-poor organizations, groups or individuals except the government that is united through various relationships (such as administrative, hierarchical, geopolitical and blood ties).

Besides, based on whether the social assistance subjects are located in rural areas, they are divided into two categories: external and internal social assistance subjects. Among them, external social assistance subjects mainly refer to those social groups or individuals who don't reside in the village, such as subordinate governmental institutions, public institutions (such as schools, hospitals and banks), colleges and universities, associations and enterprises. Internal social assistance subjects refer to party cadres of the village (such as village leaders, retired party members, members of village party and administrative committees, and college-graduate village officials), capable households (including those in charge of agriculture-related enterprises) and cooperatives (such as special planting and breeding) and other organizations or individuals in the village.

### 3.2 Empirical expectations

There are significant differences in the functions of different pairing bodies in poverty alleviation practices, including two points: the poverty alleviation work system shows a tree structure of trunk plus side branches, with apparent primary and secondary relationships. The trunk refers to the poverty alleviation system at the central, provincial, municipal, and county levels (Yuan, 2016) [21], in which the county poverty alleviation offices are mostly the frontline institutions for the poor groups and are responsible for the top-down implementation of lots of funds, projects, authority (such as audit, review, and acceptance), tasks, and responsibilities. The branch refers to the diversified socially paired assistance subjects. Social paired assistance subjects transfer existing resources to the poverty alleviation target through technical guidance, mutual help, and job provision, which enriches the scale of the team and the form of service in poverty alleviation in China and produces positive poverty alleviation results. Therefore, this paper proposes the following hypothesis:

H1. Governmental assistance subjects have a greater positive impact on poverty alleviation of paired poor households in rural China compared to social assistance subjects.

Then different social assistance subjects have advantages and vary in their strengths in helping poor households. Generally speaking, external social assistance subjects often have advantages in agricultural technology, market, information and capital, and can provide targeted, technical and professional assistance to poor households. They can help increase the income of poor households through multiple channels and means (such as labor employment recommendation, product sales and distribution, and financial credit). However, it should be noted that these measures may effectively reduce poverty in the short term. Still, they suffer from external vulnerability and instability and need to be complemented by some steps to help achieve stable poverty alleviation. In contrast, internal social assistance subjects with their more intimate social networks in rural areas have considerable advantages in identifying poor households, providing more targeted support and stimulating endogenous development momentum. Most relevant support measures are implemented through characteristic breeding industries, food-for-work and labor to promote production technology, market sharing or job provision for poor households. Therefore, their help for poor households to get rid of poverty in the short term is relatively limited. Therefore, this paper proposes the following hypothesis:

H2. External social assistance has a greater positive impact on the poverty alleviation in rural China compared to internal social assistance.

## 4. Data and methods

### 4.1 Data

This paper uses data from the information disclosure directory/annual report on the official website of the People's Government of W County, Lu'an City, Anhui Province in China and the poverty alleviation statistical reports published, disclosed and archived on the website of the *County Poverty Alleviation and Development Office* in B Town from 2016 to 2019. The raw data and workflow records are summarized and organized according to projects, categories and batches.

Some samples, such as underage heads of households formed by exceptional circumstances, are deleted to screen a total of B Township 2031 registered poor households with documented poverty (identified in 2016), basically covering all reported poor homes in the township, forming a total of 4 periods of panel data. The relevant data record information on family

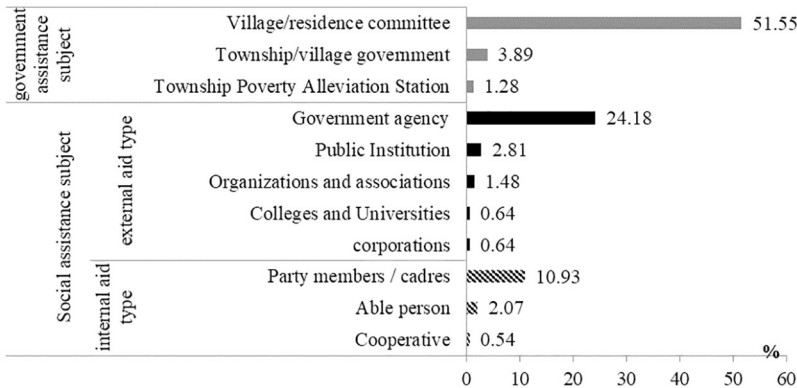

**Fig 2. Participation ratio of different paired assistance subjects.**

characteristics, poverty status, assistance measures and twinning assistance units and responsible persons of poor households from 2016 to 2019.

In terms of different types of paired assistance subjects (see Fig 2), governmental assistance subjects mainly consist of (county/district) town governments, poverty alleviation workstations and village (neighborhood) committees have helped 56.72% of the total households, and village (neighborhood) committees are the predominant supporting force, mainly because village (neighborhood) committees are generally the terminal verification units of funds, projects and market information for county-level poverty alleviation offices, serving as a connecting bond between poor households and higher-level poverty alleviation agencies, and need to complete detailed daily tasks such as data collection, classification statistics, poverty alleviation project implementation, effectiveness supervision and willingness feedback, which is like the capillaries in the poverty alleviation work network. Meanwhile, in the whole observation sample, the total proportion of households with various types of social assistance subjects is 43.28%, which indicates that social forces, through auxiliary, are effectively involved in the current poverty alleviation work and play an increasingly important role, reflecting the institutional advantages in poverty governance in China.

Furthermore, analyzing the composition of the leading social assistance subjects, we find that 29.69% and 13.59% of external and internal social assistance subjects participate in, respectively, indicating that external assistance subjects are the main component of social paired assistance subjects at this stage, which is related to the political response, economic conditions and support of associated institutions, organizations, groups or units. However, internal social assistance subject is a social support force that shouldn't be underestimated. Compared with government-affiliated institutions or teams and research institutes, individuals or small-scale organizations such as large breeders, village party members/cadres and cooperatives have mainly limited capacity, but with the cooperation and support of relevant policies and funds, and then combined with the advantages of their industries, (particular breeding) technology and market information, they are often able to lead poor households to engage in production and business activities by setting up examples, which is conducive to cultivating relatively stable and lasting means or skills for poor households to increase their income.

Next, Table 1 shows the poverty alleviation effects of poor households with different paired assistance subjects. In terms of the cumulative poverty alleviation rate of poor households from 2017 to 2019, the poverty alleviation rate of social assistance subject is slightly higher than that of governmental assistance subject, and the statistics show that by 2019, the cumulative poverty alleviation rates of poor households with social assistance and governmental

Table 1. Statistics of the poverty alleviation effects of different assistance subjects.

| | Total households | The annual number of households progressively out of poverty (%) | | | Cumulative number of households out of poverty in 2019 for different types of poor households | | |
|---|---|---|---|---|---|---|---|
| | | 2017 | 2018 | 2019 | General poor households | Low-income households | Five-guarantee households |
| Governmental assistance | 1152 | 243 (21.09) | 497 (43.14) | 889 (77.17) | 334(366) | 283(322) | 272(464) |
| Social assistance | 879 | 198 (22.53) | 453 (51.54) | 794 (90.33) | 321(350) | 242(271) | 231(258) |
| internal assistance | 276 | 51(18.48) | 74(26.81) | 213 (77.17) | 55(80) | 57(78) | 101(118) |
| external assistance | 603 | 147 (24.38) | 379 (62.85) | 581 (96.35) | 266(270) | 185(193) | 130(140) |

Note: The value in parentheses after the annual cumulative number of households out of poverty is the proportion of the number of households out of poverty to the corresponding leading group of assistance; the value in parentheses after the cumulative number of households out of poverty in 2019 for different types of poor households is the total number of households of that type, while the difference between the two is the number of households under that type of poor households that have not yet got out of poverty by 2019.

assistance are 90.33% and 77.17% respectively, and threre are two reason for this difference: first, the number of poor households with governmental assistance is about 30% more than that with social assistance; Second, the proportion of the five-guarantees households with governmental assistance is high, and the cumulative number of households out of poverty in 2019 for different types of poor households reveals that the group of five-guarantees households accounts for 40.28% of the number of households with governmental assistance, and the proportion of households among the five-guarantees households that have not yet escaped from poverty by 2019 is still 41.38%. In poverty alleviation, the governmental assistance subject focus on the five-guarantee households, reflecting a more robust social security function to provide essential livelihood, which mainly because the five-guarantee households often find it more challenging to get rid of poverty. In addition to the traditional cash consolation, material gifts and other transfer payments and subsidies, the paired assistance need to provide more assistance to help people steadily to shake off the poverty. The mobilization and operation of relevant resources still depend on the government to a certain extent.

Accordingly, the proportion of households paired with poor households for social support decreases according to general poor households, low-income households and five-guarantee households. The number of paired households is more profit to available poor households, accounting for 39.82% of the total number of households. Compared with rural low-income households and five-guarantee households, available poor households tend to have a higher ability and willingness to fight against poverty. The channels and means of social forces can help them eliminate poverty more effectively. In a comprehensive view, the participation of governmental and social parties in paired assistance play primary and secondary roles and accomplish complex and easy tasks, respectively; that is, governmental paired assistance plays a predominant role in eliminating poverty for more disadvantageous poor groups, and social paired assistance mostly play a supplementary role.

In terms of the structure of the two types of social assistance subjects, statistics show that by 2019, the cumulative poverty alleviation rates of poor households with external and internal social assistance subjects were 96.35% and 77.17%, respectively. The effect of external social assistance subjects on poverty alleviation of poor households was higher than that of internal ones, possibly because the objects of internal social assistance subjects had a higher proportion of the five-guarantee households. In contrast, the things of external social assistance subjects

mainly belonged to generally poor households. Second, the above differences reveal to some extent the effectiveness of guiding social forces such as relevant government agencies, enterprises and institutions and groups to participate in poverty alleviation, reflecting the role of superior resources for poverty alleviation and development such as technology, information and market (product sales, priority procurement) of this part of social assistance subjects.

## 4.2 Methods

The econometric model setting consists of two parts. First, based on the data format, a panel selection model is set to estimate the poverty alleviation effects of different paired assistance subjects respectively, and the specific model setting is as follows:

$$y_{it}^* = \alpha + \beta_1 HS_{it} + \sum_{k \geq 2} \beta_k \cdot Control_{kit} + v_{it} + u_{it}$$

$$y_{it} = \begin{cases} 1, & \text{if } y_{it}^* > 0 \\ 0, & \text{if } y_{it}^* \leq 0 \end{cases}$$

(1)

In Eq (1), $y_{it}^*$ is an unobservable latent variable; $y_{it}$ is the dependent variable, indicating the poverty status of the $i$-th documented poor household in a the $t$-the year, with 0 and 1 indicating poverty and escape from poverty, respectively. The main criterion to judge whether a poor household is out of poverty is that the net per capita income of the household steadily exceeds the national poverty alleviation standard (The current poverty line is based on the constant price of 2,300 yuan in 2011 and is fine-tuned yearly according to the price index. 2016 poverty line is about 3,000 yuan, the 2017 poverty line is about 3,300, and the 2018 poverty line is about 3,400.) of that year. Denotes the type of poor households' pairing subjects, with 0 and 1 denoting governmental and social assistance subjects, respectively. Indicates $k$ control variables affecting poor households, including household characteristics, external environment characteristics and type of poor households (According to the requirements of the National Poverty Alleviation Office, the types of poor households include general poor households, low-income poor households and five-guarantee poor households, among which low-income poor households and five-guarantee poor households enjoy low-income subsidies and five-guarantee subsidies respectively.). And denote the corresponding coefficients to be estimated; the residual term is by a normal distribution.

Second, considering that the participation of different assistance subjects in paired support is a vital poverty alleviation policy put into practice, most methods based on mean comparison or multiple regressions do not quickly solve the difficulties of endogeneity (including associativity, omitted variables or measurement errors) and causality identification, so this paper uses PSM-DID method to solve the problem of selectivity bias. The main reason for using this method is that the paired relationship between poor households and the assistance subjects (such as organizations, institutions, enterprises or individuals) are not familiar with each other before the paired relationship is established. The paired relationship is "arrange the coordination of the county poverty alleviation office resources. The randomness of the above-paired relationship can eliminate the selective bias, which is more suitable for identifying the causal relationship between policy and behavior. The double difference estimator (DID) is calculated as follows:

$$DID = [E(y_{it}|T_{it} = 1, treated_i = 1) - E(y_{it}|T_{it} = 0, treated_i = 1)]$$
$$- [E(y_{it}|T_{it} = 1, treated_i = 0) - E(y_{it}|T_{it} = 0, treated_i = 0)]$$

(2)

The processing of Eq (2) is divided into two steps. The first step is propensity score matching. Before matching, poor households with governmental support were identified as the treatment group and poor households with social support as the control group. Then six observation variables, identified as a poor village, far away from the urban area, arable land per capita, Whether it is plain terrain, type of poor household, education of household head, number of the household laborers and gender of household head, were selected for caliper matching for the treatment and control groups respectively. The index meanings and units of specific variables are shown in Table 2.

The second step is the double difference estimation. For the treatment group obtained after the PSM process, the dummy variable *treated = 1* was set, while the control group set dummy variables *treated = 0*. Similarly, the time dummy variable was set $T$. Since the data collection was started in 2016, and there is a time lag between the implementation of poverty alleviation measures and the presentation of their effects, i.e., the results of the assistance measures implemented for poor households in 2016 can only be entirely determined in early 2017 at the earliest, so the time dummy variable for 2016 is and that for subsequent years is $T = 1$.

The following double difference model is developed to observe the poverty alleviation effects and differences between governmental and social assistance for poor households.

$$y_{it} = \lambda_0 + \lambda_1 treated_i + \lambda_2 T_i + \lambda_3 treated_i \times T_i + \delta_k Control_{kit} + \mu_t + \varepsilon_{it} \tag{3}$$

In this Eq (3), for the control group of social assistance subjects (*treated = 0*), the poverty status of farm households before the implementation of the assistance policy is $\lambda_0$, the poverty status of farm households after the performance is $\lambda_0 + \lambda_2$, and the difference in the poverty status of the control group before and after the assistance policy is $dif_0 = \lambda_2$; for the treatment group, the poverty status of farm households before the implementation of the assistance policy is $\lambda_0 + \lambda_1$, and the poverty status of farm households after the performance is $\lambda_0 + \lambda_1 + \lambda_2 + \lambda_3$, the difference in the poverty status of the treatment group before and after the assistance policy was $dif_1 = \lambda_2 + \lambda_3$, and the net effect of implementing the assistance policy in the poverty status of farm households is $DID = dif_1 - dif_0 = \lambda_3$.

**Table 2. Definition of indicators for explanatory variables.**

| Variables | Definition (unit) |
|---|---|
| Core explanatory variables | |
| Governmental assistance | The main attribute of subjects paired with poor households: governmental assistance = 1, social assistance = 0 |
| Social assistance | Type of social support subjects paired with poor households: external assistance = 1, internal assistance = 0 |
| Control variables | |
| Identified as a poor village | The village where the poor household is located is identified as a poor village: Yes = 1, No = 0. |
| Distance to the city | Distance of road between the village of the poor household and the city (km) |
| arable land per capita | Statistics of the per capita area of arable land |
| Whether it is plain terrain | Plain = 0, other = 1 |
| Type of poor household | Identification of the type of poor households registered in the data: general poor households = 0, low-income households or Five-guarantees households = 1 |
| Education of the head of household | Educational attainment of the head of household: college and above = 1, high school, secondary school = 2, junior high school = 3, primary school = 4, no schooling = 5 |
| Number of household laborers | Number of laborers (aged 16–59) in poor households (persons) |
| Gender of the head of household | Gender of the head of poor households registered: female = 1, male = 0. |

Moreover, the impact of internal and external social assistance subjects on the poverty alleviation of poor households was analyzed with helped poor households as a sample, mainly following the same treatment steps described above, where the external social assistance subject was set as the treatment group. In contrast, the internal social assistance subject was the control group.

## 5. Analysis of empirical results

### 5.1 The impact of different assistance subjects to poverty alleviation of paired poor households

Since some of the control variables in the data do not change over time, the panel fixed effect model assumes that these non-time invariant control variables will not affect dependent variables. Thus, the article uses a panel random-effects model to analyze the poverty alleviation effects of different assistance subjects. In Table 3, column (1) reports only the panel selection model (Logit) of the estimated results of governmental assistance on farmers' poverty alleviation, while column (2) reports the degree of influence of governmental assistance and control variables on farmers' poverty alleviation. Additionally, columns (3) and (4) both use a subsample of socially assisted poor households to report the poverty alleviation effects of external social assistance versus internal social assistance, where column (3) reports only the extent of the impact of external social assistance on farmers' poverty alleviation and column (4) reports the time of the effect of external social assistance and control variables on farmers' poverty alleviation.

According to the model estimated results in Table 3, it is found that the effect of governmental assistance on poverty alleviation of paired poor households is significantly negative compared to social support, i.e., social support has a significant contribution to poverty alleviation of paired poor households, but it is also noted that the parameter estimation of governmental assistance in column (2) after adding relevant control variables are significantly lower than those in column (1), indicating that a single view of governmental assistance on poverty alleviation of paired poor households may be affected by other factors and may be overestimated to some extent. The extent of government assistance may be influenced by other factors and may be overestimated. Similarly, in the subsample of social assistance for poor households, the effect of external social assistance on poverty alleviation of paired poor households in

**Table 3. The panel selection model of different assistance subjects' estimated effects of poverty alleviation on poor households.**

|  | (1) | (2) | (3) | (4) |
|---|---|---|---|---|
| Governmental assistance subjects | -0.0575***(0.0113) | -0.0283***(0.0108) |  |  |
| social assistance subjects |  |  | 0.1456***(0.0158) | 0.0349*(0.0193) |
| Identified as a poor village |  | 0.2159***(0.0190) |  | 0.130***(0.0278) |
| Distance to the city |  | 0.0027*(0.0015) |  | 0.0110***(0.0020) |
| arable land per capita |  | 0.2341***(0.0587) |  | 0.3023***(0.0906) |
| Whether it is plain terrain |  | -0.2179***(0.0544) |  | -0.2287***(0.0806) |
| Type of poor household |  | -0.1146***(0.0120) |  | -0.0967***(0.0174) |
| Education of the head of household |  | 0.0126(0.0086) |  | 0.0324**(0.0129) |
| Number of household laborers |  | 0.0358***(0.0059) |  | 0.0252***(0.0085) |
| Gender of the head of household |  | 0.0113(0.0159) |  | -0.0129(0.0240) |
| Number of samples | 8124 | 8124 | 3516 | 3516 |
| $Waldchi2(N)$ | 25.74*** | 512.91*** | 77.79*** | 255.10*** |

**Note**: (1) ***, **, *indicate significance at the 1%, 5% and 10% levels, respectively. (2) The reporting parameter is the marginal effect.

column (3) is significantly positive, indicating that the impact of external social assistance on poverty alleviation is better than that of internal social assistance. However, the significance of this parameter changes in column (4) after adding control variables, and the relevant reasons are worth tracing.

## 5.2 The impact of governmental assistance to poverty alleviation of paired poor households

Propensity value matching (PSM) was conducted for the paired households with governmental and social assistance. Fig 3 depicts the kernel density function curves of the treatment and

**(a)**

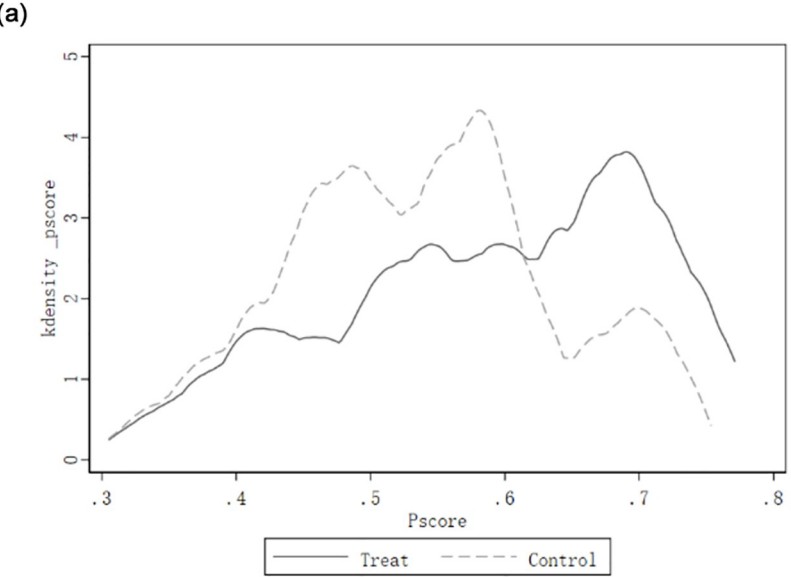

**(b)**

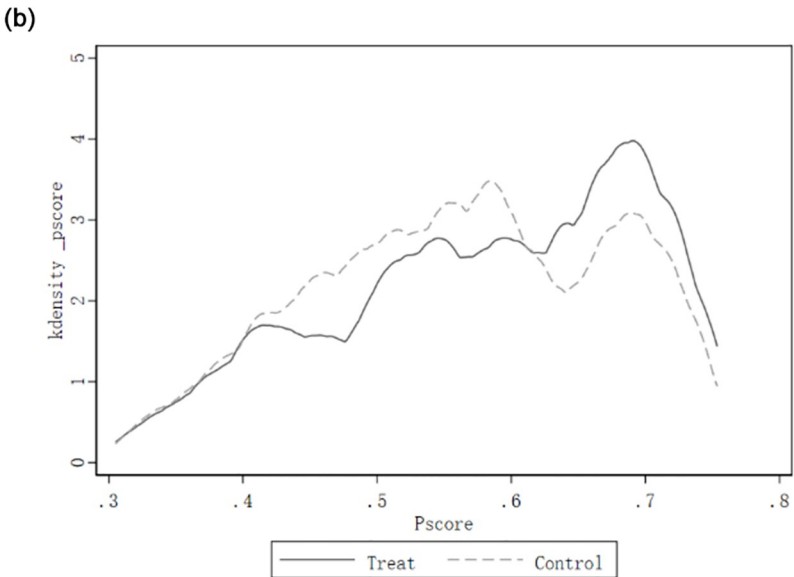

**Fig 3. Comparison of the distribution of kernel density of propensity score values in the treatment and control groups before and after total sample caliper propensity matching.** a. Kernel density of pre-match propensity scores. b. Kernel density of post-match propensity scores.

**Table 4. DID and PSM-DID estimation results of the impact of governmental assistance on poverty alleviation of poor households.**

|  | DID | | PSM-DID | |
|---|---|---|---|---|
|  | (1) | (2) | (3) | (4) |
| Governmental assistance subjects × T | -0.0541***(0.0205) | -0.0819***(0.0209) | -0.0507**(0.0206) | -0.0780***(0.0208) |
| Governmental assistance subjects | -0.0158(0.0191) | 0.0328*(0.0186) | -0.0134(0.0191) | 0.0320*(0.0186) |
| T | 0.4790***(0.0131) | 0.5017***(0.0136) | 0.4760***(0.0129) | 0.4980***(0.0135) |
| Identified as a poor village |  | 0.2210***(0.0182) |  | 0.2223***(0.0181) |
| Distance to the city |  | 0.0035**(0.0014) |  | 0.0039***(0.0014) |
| arable land per capita |  | 0.2381***(0.0552) |  | 0.2714***(0.0560) |
| Whether it is plain terrain |  | -0.2217***(0.0513) |  | -0.2466***(0.0516) |
| Type of poor household |  | -0.1198***(0.0115) |  | -0.1182***(0.0114) |
| Education of the head of household |  | 0.0130(0.0082) |  | 0.0175**(0.0083) |
| Number of household laborers |  | 0.0348***(0.0056) |  | 0.0345***(0.0056) |
| Gender of the head of household |  | 0.0108(0.0151) |  | 0.0118(0.0152) |
| Number of samples | 8124 | 8124 | 7964 | 7964 |
| Waldchi2(N) | 1239.27*** | 1327.29*** | 1218.21*** | 1302.56*** |

Note: (1) ***, **, * indicate significance at the 1%, 5% and 10% levels, respectively. (2) The reporting parameter is the marginal effect. (3) 160 unmatched data are deleted after propensity matching, so the sample size of the PSM-DID model is reduced.

control groups before and after the caliper propensity score matching. It is easy to find that before matching, the probability density distributions of the propensity score values of the two groups of samples were significantly different. The bias in the treatment group was more pronounced (see Fig 3a). However, after finishing caliper matching, the probability density distributions of the retained two groups of samples converged (see Fig 3b), indicating that the characteristics of the two groups of samples after matching were very close in all aspects. The selective bias of the models was eliminated. In addition, the equilibrium test of the matched covariates found that the "standardized gap" of all control variables after checking was significantly less than 10%, indicating that there was no systematic difference between the propensity scores of the two groups after matching, and the requirement of comparability was met.

Table 4 reports the degree of impact of governmental assistance on poverty alleviation of paired poor households, where column (1) is the total sample DID model with governmental assistance and time only, and column (2) is the total sample DID model with governmental assistance and time and control variables; columns (3) and (4) are the samples after using propensity value matching and report the estimated results of the PSM-DID model with governmental assistance help and time and control variables.

The results of the interaction term between governmental assistance and time in columns (1)-(4) in Table 4, with or without the addition of control variables, show that governmental assistance has a significant negative effect on poverty alleviation among paired poor households, where the estimated results in column (4) indicate that the net impact of governmental assistance on poverty alleviation among paired poor households is -0.078 and hypothesis 1 is not verified. This parameter indicates that the participation of social forces in paired assistance can significantly contribute to a 7.8 percentage point increase in the poverty eradication rate of poor households compared to governmental assistance. The difference in poverty eradication rates between the above two types of assistance subjects for paired poor households is mainly attributed to the significant difference in the types of paired poor households, especially the reality that the difficulty of poverty eradication is significantly more meaningful for the low-income and five-guarantee households than the general poor households (Luo, 2022) [22]. The

governmental assistance subjects mainly take up this part of the "relatively difficult" poverty eradication group.

### 5.3 The impact of internal and external social assistance subjects to poor households' poverty alleviation

Furthermore, the article examines the mechanisms and causes of differences in poverty alleviation by social assistance subjects in paired poor households, using a subsample of social assistance subjects in paired poor households to explore the effects of two types of social assistance subjects, external and internal, on poverty alleviation among poor households. To reduce the selectivity deviation of the estimated model, the nearest neighbor propensity value matching was conducted for the two groups of external and internal assistance subjects while keeping the sample intact as much as possible. Table 4 reports the kernel density plots of the two groups before and after matching and finds that the probability density distribution of the retained two groups of samples tends to be consistent (see Fig 4a and 4b), indicating that the characteristics of the two groups of samples after matching have been very close to each other and the matching results are excellent. In addition, the balance test of the models showed that the corresponding values of each covariate were significantly less than 10%, and the matching results passed the balance test.

Table 5 reports the degree of impact of external social assistance on poverty alleviation of paired poor households, where column (1) is the total sample DID model with only external social assistance and time $T$, and column (2) is the total sample DID model with external social assistance and time and control variables; columns (3) and (4) are the estimated results of the PSM-DID model with external social assistance and time and control variables using propensity value matching post-sample.

In Table 5, regardless of whether control variables are added or not, the results of the interaction between external social support and time in columns (1)-(4) show that external social support has a significant positive effect on poverty alleviation among paired poor households, e.g., the estimated result in column (4) shows that the net impact of external social support on poverty alleviation this significant poverty alleviation effect is mainly due to the economic base and poverty alleviation efforts "carried" by external social assistance subjects, among paired poor households is 0.1426, and Hypothesis 2 is verified. This parameter result indicates that external social assistance subjects can significantly increase the poverty alleviation rate of poor households by 14.26 percent, including financial consolation, agricultural products marketing, job creation and microcredit.

At the same time, internal social assistance does not significantly impact the poverty alleviation rate of the paired poor households compared to external social assistance. Therefore, the principle of moderation should be advocated for constructing stable internal social support forces in the future, together with corresponding incentives or support measures.

## 6. Further discussion

### 6.1 Heterogeneity analysis from the perspective of poor household types

In both the statistical description of the sample and the empirical analysis sections, explaining the poverty alleviation effects of governmental assistance and external social assistance on paired poor households involves discussing the types of poor households. It is found that the differences in the types of poor households affect the rigor of the conclusions to some extent. Therefore, in this section, the total sample is separated into three subsamples according to the type of poor households. The poverty alleviation effect of governmental assistance on paired

**(a)**

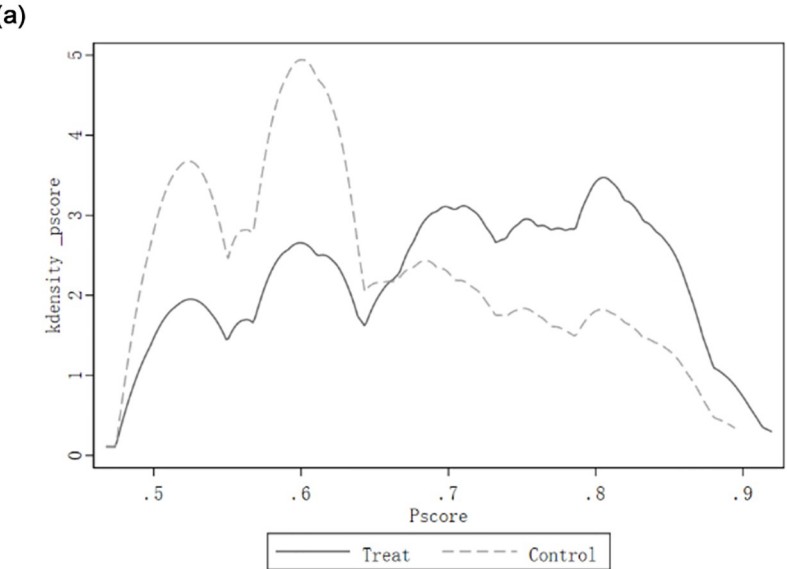

**(b)**

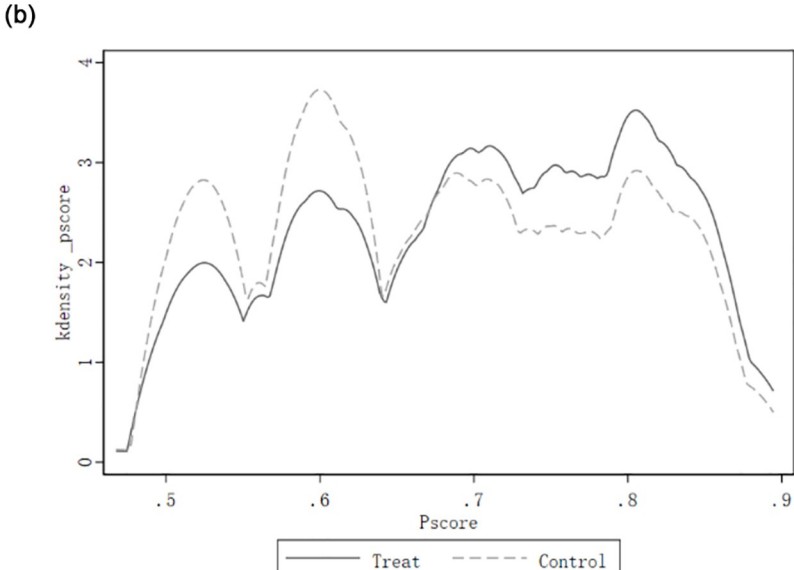

**Fig 4. Comparison of the kernel density distribution of propensity score values in the treatment and control groups before and after nearest neighbor propensity matching.** a. Kernel density of pre-match propensity scores. b. Kernel density of post-match propensity scores.

poor households is re-estimated for each subsample using a propensity matching difference model. The relevant model results are shown in columns (1)-(3) in Table 6.

Furthermore, the sample of poor households paired with social support is split into three subsamples according to the type of poor households. The propensity matching difference model is used for each subsample to estimate the poverty alleviation effect of external social support on poor households. The related model results are shown in columns (4)-(6) in Table 6.

Comparing the results of the interaction term between governmental assistance and time $T$ in columns (4) in Table 4 and (1)-(3) in Table 6, it is found that the differential results of this

**Table 5. DID and PSM-DID estimation results of the impact of external social pairs assistance on poverty alleviation of poor households.**

| | DID | | PSM-DID | |
|---|---|---|---|---|
| | **(1)** | **(2)** | **(3)** | **(4)** |
| social assistance subjects × T | 0.1166***(0.0306) | 0.1415***(0.0307) | 0.1173***(0.0308) | 0.1426***(0.0309) |
| social assistance subjects | 0.0536*(0.0281) | -0.0805***(0.0295) | 0.0519*(0.0283) | -0.0821***(0.0297) |
| T | 0.3562***(0.02341) | 0.3503***(0.0230) | 0.3570***(0.0235) | 0.3510***(0.0231) |
| identified as a poor village | | 0.1362***(0.0247) | | 0.1360***(0.0248) |
| Distance to the city | | 0.0119***(0.0018) | | 0.0121***(0.0018) |
| arable land per capita | | 0.3174***(0.0808) | | 0.3144***(0.0811) |
| Whether it is plain terrain | | -0.2337***(0.0712) | | -0.2338***(0.0715) |
| Type of poor household | | -0.1028***(0.0153) | | -0.1024***(0.0154) |
| Education of the head of household | | 0.0326***(0.0111) | | 0.0329***(0.0112) |
| Number of household laborers | | 0.02523***(0.0074) | | 0.0239***(0.0075) |
| Gender of the head of household | | -0.0150(0.0208) | | -0.0169(0.0212) |
| Number of samples | 3516 | 3516 | 3488 | 3488 |
| Waldchi2(N) | 585.64*** | 604.49*** | 582.10*** | 600.89*** |

**Note**: (1) ***, **, * indicate significance at the 1%, 5% and 10% levels, respectively. (2) The reporting parameter is the marginal effect. (3) 28 unmatched data are deleted after propensity matching, so the sample size of the PSM-DID model is reduced.

parameter estimated values remain consistent after matching the total sample and matching the subsamples split according to the type of poor households, and governmental assistance still has a significant negative effect on poverty elimination for different types of paired poor households, but the observed samples in columns (1)-(3) in Table 6 are increasing as the poor households evolve from The absolute value of this parameter value is growing as the type of poor households evolves from general poor households, low-income households to five-guarantee households, and this progression of parameter values reflects the pattern of increasing difficulty for governmental assistance to get different types of poor households out of poverty,

**Table 6. PSM-DID estimate the poverty alleviation effect of different assistance subjects based on the perspective of poor household types.**

| | General poor households | Low-income households | Five-guarantee households | General poor households | Low-income households | Five-guarantee households |
|---|---|---|---|---|---|---|
| | **(1)** | **(2)** | **(3)** | **(4)** | **(5)** | **(6)** |
| Governmental assistance subjects × T | -0.0188(0.3784) | -0.1448***(0.0502) | -0.1252***(0.0434) | | | |
| Governmental assistance subjects | 0.0029(0.0248) | 0.1507***(0.0477) | -0.0012(0.0396) | | | |
| social assistance subjects × T | | | | 0.1273***(0.0420) | 0.2484***(0.0774) | 0.1007(0.0773) |
| social assistance subjects | | | | -0.0231(0.0423) | -0.1486*(0.0766) | -0.1658**(0.0735) |
| T | 0.3784***(0.0177) | 0.6433***(0.0377) | 0.5206***(0.0314) | 0.2528***(0.0338) | 0.4108***(0.0559) | 0.4415***(0.0413) |
| Control variables | Control | Control | Control | Control | Control | Control |
| Number of samples | 2852 | 2368 | 2852 | 1368 | 1080 | 892 |
| Waldchi2(N) | 371.39*** | 534.43*** | 451.49*** | 173.49*** | 241.85*** | 165.18*** |

**Note**: (1) Estimates of control variables are not reported due to space limitations; (2) Reported parameters are marginal effects; (3) ***, **, and * indicate significance at the 1%, 5%, and 10% levels respectively; (4) Values in parentheses in the three columns on the right in Table 2 can be summed to obtain the total number of poor households of different types. The sample size of the poor households corresponding to each column in this table is smaller than the above total number of households×4, mainly because some samples were omitted in the matching.

i.e., as the difficulty of poverty eradication of poor households from general poor households, low-income households to five-guarantee households increases in reality, the effect of governmental assistance on poverty eradication of paired poor households decreases correspondingly.

Correspondingly, the estimated values of the interaction term between external social assistance subjects and time in columns (4) in Table 5 and (4)-(6) in Table 6 remain consistent with the different results after matching the social assistance subsample and matching the subsamples separated again according to the type of poor households, indicating that external social assistance has a significant positive effect on poverty alleviation for different types of paired poor households. However, it is worth noting that the estimated levels of the interaction term between external social assistance and time are significantly pulled down in the subsample of five-guarantee households, highlighting the enormity of poverty alleviation for the group of five-guarantee households in the precision poverty alleviation work.

## 6.2 Robustness tests

The parameter estimation of the previous panel selection model and PSM-DID model are Logit models. To avoid the potential influence of the selection of different distribution functions on the results in the selection model, the Probit model is used for robustness testing.

Comparing Tables 4 and 5 shows that the parameter estimation of the impact of governmental assistance and external social assistance on poverty alleviation of paired poor households in Table 7 remains consistent and stable, indicating the relative robustness of the empirical results of the article.

Furthermore, the earlier part of this essay used caliper matching for the total sample and nearest-neighbor matching for the social assistance subsample. To avoid the possible influence of choosing a single matching method on the measurement results, Table 8 reports the average treatment effects (ATT) obtained by taking five standard matching methods: nearest-neighbor matching, nearest-neighbor matching within the caliper, caliper matching, kernel matching and spline matching for the total sample and social assistance subsample respectively. The relevant parameters show that the average treatment effect values of the full model and the social assistance subsample under different matching methods remain relatively stable within the range of variation, indicating that the matching methods chosen in the article are appropriate. At the same time, the research results are pretty robust and reliable.

**Table 7. Estimated results of the impact of different assistance subjects on the poverty alleviation of paired poor households (Probit model).**

|  | DID | PSM-DID | DID | PSM-DID |
|---|---|---|---|---|
|  | (1) | (2) | (3) | (4) |
| Governmental assistance subjects $\times T$ | -0.0762***(0.0205) | -0.0725***(0.0205) |  |  |
| Governmental assistance subjects | 0.0285(0.0185) | 0.0279(0.0132) |  |  |
| social assistance subjects $\times T$ |  |  | 0.1382***(0.0304) | 0.1391***(0.0306 |
| social assistance subjects |  |  | -0.0784***(0.0292) | -0.0800***(0.0294) |
| $T$ | 0.5071***(0.0133) | 0.5040***(0.0132) | 0.3629***(0.0230) | 0.3638***(0.0230) |
| Control variables | Control | Control | Control | Control |
| Number of samples | 8124 | 7964 | 3516 | 3488 |
| Waldchi2(N) | 1538.69*** | 1511.87*** | 709.35*** | 705.05* |

**Note**: (1) Estimates of control variables are not reported due to space limitations; (2) Reported parameters are marginal effects; (3) ***, **, and * denote significance at the 1%, 5%, and 10% levels respectively; (4) Sample size for the PSM-DID model is reduced due to the removal of unmatched data.

**Table 8. Average treatment effects under different matching methods for the total sample, social support sub-sample.**

| PSM | Nearest neighbor matching | Nearest-neighbor matching within the caliper | Caliper matching | Kernel matching | Spline matching |
|---|---|---|---|---|---|
| Governmental assistance subjects (total sample) | -0.0449**(0.0214) | -0.0449**(0.0214) | -0.0336*** (0.0113) | -0.0311*** (0.0119) | -0.0283** (0.0124) |
| social assistance subjects(sub-sample) | 0.1245***(0.0218) | 0.1245***(0.0218) | 0.1434*** (0.0198) | 0.1449*** (0.0195) | 0.1447*** (0.0190) |

**Note**:

***, **, and * denote significance at the 1%, 5%, and 10% levels respectively.

## 7. Conclusion

The empirical results demonstrate that compared to government assistance, participation of social forces in paired poverty alleviation has more significant positive effects on poverty reduction for paired households. After sample matching and controlling for relevant variables, it is found that involvement of social forces in paired poverty alleviation can markedly increase the poverty reduction rate of poor households by 7.8 percentage points. Additionally, further analysis of the subsample of households receiving assistance from social forces using the PSM-DID model indicates that externally-introduced social assistance subjects can significantly improve the poverty reduction rate of poor households by 14.26 percentage points. This pronounced poverty alleviation effect is primarily attributed to the robust economic foundations and sustained poverty reduction efforts of external social assistance providers. In summary, the empirical evidence affirms that engagement of social forces, especially external ones, in poverty alleviation pairing assistance is more impactful for improving the poverty reduction outcomes of assisted households compared to government assistance.

The results of this study make important contributions to sustainable development theory, especially in emphasizing the role of social forces in integrating poverty alleviation resources and pursuing sustainable development goals. By reducing poverty rates, social forces not only address short-term economic needs, but also contribute to long-term sustainability. This means that the participation of social forces not only has a positive impact on the current economic situation, but also plays an important role in long-term sustainable development. This provides an empirical support for sustainable development theory, demonstrating that poverty reduction strategies must be designed and implemented in a manner that takes into account their impact on sustainable development and incorporates sustainable development theory. The research results also provide insights for the new public management theory. The significant impact of paired assistance on poverty reduction highlights the importance of policy decisions in mobilizing social participation in poverty reduction. These conclusions can provide a practical basis for encouraging and promoting the participation of social forces in future poverty reduction policies.

In addition, this paper argues that to build a sustainable poverty alleviation ecosystem in the future, we should insist on mobilizing social forces to participate in poverty alleviation and development under the leadership of government help, and optimize the system of foreign aid-type social twinning help subjects to deeply participate in poverty governance and refine innovative help measures. Specifically, these include:

Firstly, the considerable endeavors exerted by the government in their poverty alleviation campaign cannot be casually negated. This recognition arises from the fact that, in the course of providing assistance, the government, in addition to assuming responsibility for an increased number of five-guarantee households, is also obligated to undertake a sequence of

infrastructural developments. These substantial undertakings invariably necessitate an extensive commitment of both human capital and fiscal resources. Second, the concept of encouraging and guiding social forces to participate in poverty management should be continuously implemented. The integration of various types of social forces into poverty alleviation and development work enriches and improves the efficiency of resource allocation of human, material and financial resources in poverty alleviation, and greatly complements and promotes the completion of poverty alleviation work. The addition of social forces not only relieves the pressure of government help, but also makes the help work more precise and efficient.

Third, optimizing institutional safeguards is imperative to incentivize greater engagement of external social assistance actors in targeted poverty alleviation initiatives. The information, technology, and market advantages of external social pairing helpers are scarce resources in poverty alleviation and development, and to attract more external helpers to join in poverty alleviation work, it is also necessary to improve the relationship between the relevant organizations and build a favorable working environment.

Fourth, internal social assistance should not be ignored. Subjects must be helped to not only to earn income, but also to stimulate their internal motivation, mobilize their enthusiasm, initiative and creativity, and form an interactive mechanism between external diversified poverty alleviation and internal self-poverty alleviation.

## Supporting information

**S1 Data.**
(CSV)

## Author Contributions

**Conceptualization:** Sai Zhu.

**Data curation:** Zhongbao Tian, Sai Zhu.

**Formal analysis:** Zhongbao Tian.

**Writing – original draft:** Quanzhong Wang, Zhongbao Tian.

**Writing – review & editing:** Quanzhong Wang, Zhongbao Tian.

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
