## [Decision Letter · Decision Letter 0]

18 Sep 2023

PONE-D-23-23611Paired assistance and poverty alleviation: Experience and evidence from ChinaPLOS ONE

Dear Dr. Wang,

Thank you for submitting your manuscript to PLOS ONE. After careful consideration, we feel that it has merit but does not fully meet PLOS ONE’s publication criteria as it currently stands. Therefore, we invite you to submit a revised version of the manuscript that addresses the points raised during the review process.

We look forward to receiving your revised manuscript.

Kind regards,

Kashif Abbass, Ph.D

Academic Editor

PLOS ONE

Journal Requirements:

"This research was funded by the Anhui Philosophy and Social Science Planning Project: “Research on Poverty Reduction Mechanism of Supported Industries from the Perspective of double-win for Villages and Households” (grant number AHSKQ2019D112)."

"NO authors have competing interests"

Reviewers' comments:

Reviewer's Responses to Questions

**Comments to the Author**

1. Is the manuscript technically sound, and do the data support the conclusions?

Reviewer #1: Partly

Reviewer #2: Yes

2. Has the statistical analysis been performed appropriately and rigorously? 

Reviewer #1: Yes

Reviewer #2: Yes

3. Have the authors made all data underlying the findings in their manuscript fully available?

Reviewer #1: Yes

Reviewer #2: Yes

4. Is the manuscript presented in an intelligible fashion and written in standard English?

Reviewer #1: No

Reviewer #2: Yes

5. Review Comments to the Author

Reviewer #1: Authors of “Paired assistance and poverty alleviation: Experience and evidence from China” have written an interesting paper and put a lot of effort and can be accepted after major corrections. I have the following suggestions.

The current scope of the literature review is restricted in its coverage. To enhance its comprehensiveness, authors should incorporate recent advancements from the field. In the introduction, it is imperative to articulate the research questions that will guide this study.

Within the literature review section, a distinct and unequivocal hypothesis should be put forth to provide a clear direction for the research.

To improve the overall readability of the paper, it is advised to reduce the excessive use of first-person pronouns.

A comprehensive discussion of the results is needed, along with an exploration of how these findings contribute to the development of the underlying theoretical framework.

In general, it is recommended to revise the reference list to encompass the latest advancements in the literature concerning post-COVID-19 poverty alleviation. Furthermore, it is pertinent to outline potential future directions pertaining to sustainable development. To augment the reference list, the following pertinent literature should be included:

https://doi.org/10.1108/IJOEM-11-2020-1417

https://doi.org/10.1142/S021964922240010X

https://doi.org/10.1016/j.jbef.2020.100341

https://doi.org/10.1016/j.frl.2020.101640

https://doi.org/10.1007/s11356-023-26631-z

https://doi.org/10.1007/s11356-022-24458-8

https://doi.org/10.1016/j.enpol.2011.02.026

https://doi.org/10.1016/j.jclepro.2020.125143

Reviewer #2: Revision Comments

The topic is interesting but a serious level of hard work is required for acceptance and publication in PLOS-ONE.

General Comments:

It is very much astonishing for me that manuscript does not have page numbers and continuous line numbers even. How can I make a review and mention appropriate changes in the various sections of the manuscript. Note that the formatting of the manuscript highly matters for evaluation.

1.What is the background of conducting this study mention in the abstract.

2.Why the study is conducted on Anhui province of China. Is it appropriate to evaluate an entire country situation based on single province data? Justify

3.Mention practical and social implication of the study in abstract.

4.The gap of study is not clear. Mention appropriate reference of the prior literature that why that study is necessary?

5.The research questions are missing in the manuscript.

6.Elaborate the problem of your study using some graphical representations in introduction.

7.Poverty is a global issue but what is the contribution of China in reducing poverty compare with world? Draw some graphs related to this in introduction section.

8.Separate the literature review section from introduction.

Enrich the literature review by the below mentioned studies

https://link.springer.com/article/10.1007/s11356-021-17099-w

http://ramss.spcrd.org/index.php/ramss/article/view/255

9.Give the summary of literature review with appropriate research gap.

10.Separate the sections of literature review according to the linkage of variables.

11.Provide a related theocratical support for your study.

12.Methodology section missing the information and sources of research variables. Provide a table for the sources, measuring units, and descriptions of the variables.

13.Where, is the conceptual framework of the study? Where, are the research hypothesis? Provide a diagram that shows the connections of variables.

14.Give the full form of PSM-DID method.

15.You have used both primary and secondary data give logical reasons?

16.What do you think the regressed data span is sufficient for some empirical analysis?

17.Check the third mathematical equation. Also, verify all the mathematical equations in the manuscript.

18.The conclusion section needs more explanation and practical policy implications section after the conclusion.

6. PLOS authors have the option to publish the peer review history of their article (what does this mean?). If published, this will include your full peer review and any attached files.

Reviewer #1: No

Reviewer #2: No

---

## [Author Response · Author response to Decision Letter 0]

20 Dec 2023

Thanks to the suggestions of the reviewers, the suggestions of the reviewers have been revised one by one.For specific reply, please refer to the document "Response to Reviewers"

---

## [Decision Letter · Decision Letter 1]

2 Jan 2024

Paired assistance and poverty alleviation: Experience and evidence from China

PONE-D-23-23611R1

Dear Dr. Wang,

We’re pleased to inform you that your manuscript has been judged scientifically suitable for publication and will be formally accepted for publication once it meets all outstanding technical requirements.

Kind regards,

Grigorios L. Kyriakopoulos, 2 PhDs, 3 MSc, 2 MA, MEng, 2 BA, BSc

Academic Editor

PLOS ONE

Additional Editor Comments (optional):

Reviewers' comments:

Reviewer's Responses to Questions

**Comments to the Author**

1. If the authors have adequately addressed your comments raised in a previous round of review and you feel that this manuscript is now acceptable for publication, you may indicate that here to bypass the “Comments to the Author” section, enter your conflict of interest statement in the “Confidential to Editor” section, and submit your "Accept" recommendation.

Reviewer #1: All comments have been addressed

2. Is the manuscript technically sound, and do the data support the conclusions?

Reviewer #1: Yes

3. Has the statistical analysis been performed appropriately and rigorously? 

Reviewer #1: Yes

4. Have the authors made all data underlying the findings in their manuscript fully available?

Reviewer #1: No

5. Is the manuscript presented in an intelligible fashion and written in standard English?

Reviewer #1: Yes

6. Review Comments to the Author

Reviewer #1: Authors of "Paired assistance and poverty alleviation: Experience and evidence from China" have successfully done all the required changes.

7. PLOS authors have the option to publish the peer review history of their article (what does this mean?). If published, this will include your full peer review and any attached files.

Reviewer #1: No

---

## [Editor Report · Acceptance letter]

10 Feb 2024

PONE-D-23-23611R1 

PLOS ONE

Dear Dr. Wang, 

I'm pleased to inform you that your manuscript has been deemed suitable for publication in PLOS ONE. Congratulations! Your manuscript is now being handed over to our production team.

Kind regards, 

on behalf of

Dr. Grigorios L. Kyriakopoulos 

Academic Editor

PLOS ONE